# Factors Associated with Low Lean Mass in Early Rheumatoid Arthritis: A Cross-Sectional Study

**DOI:** 10.3390/medicina55110730

**Published:** 2019-11-08

**Authors:** Raili Müller, Mart Kull, Kaja Põlluste, Annika Valner, Margus Lember, Riina Kallikorm

**Affiliations:** 1Institute of Clinical Medicine, Tartu University, 50090 Tartu, Estonia; Mart.kull@gmail.com (M.K.); Kajap@ut.ee (K.P.); Annika.valner@kliinikum.ee (A.V.); Margus.lember@ut.ee (M.L.); Riina.kallikorm@ut.ee (R.K.); 2Department of Internal Medicine, Tartu University Hospital, 50406 Tartu, Estonia; 3Viljandi County Hospital, 71024 Viljandi maakond, Estonia

**Keywords:** rheumatoid arthritis, early arthritis, body composition, sarcopenia, lean mass

## Abstract

*Background and Objectives*: The aim of the study was to evaluate body composition (BC) of rheumatoid arthritis (RA) patients at disease onset compared to population controls focusing on the associations between low lean mass and disease specific parameters, nutritional factors and physical activity. *Materials and Methods*: 91 patients with early rheumatoid arthritis (ERA) (72% female) and 328 control subjects (54% female) were studied. BC-lean and fat mass parameters were measured with a Lunar Prodigy Dual Energy X-Ray Absorptiometry (DXA) machine. The prevalence, age and gender adjusted odds ratios of having low lean mass and overfat, associations between nutrition, physical activity, and ERA disease specific parameters and the presence of low lean mass were evaluated. *Results*: We found that the BC of patients with recent onset RA differs from control subjects—ERA patients had a higher mean body fat percentage (BFP) and lower appendicular lean mass (ALM). 41.8% of the ERA patients and 19.8% of the controls were classified as having low lean mass adjusted OR 3.3 (95% C.I. 1.9–5.5, *p* < 0.001). 68.1% of the ERA subjects and 47.3% of the controls were overfat (adjusted OR 1.9 (95% C.I. 1.1–3.3, *p* = 0.02)) and the adjusted odds of having both low lean mass and overfat were 4.4 times higher (26.4% vs. 7.0% 95% C.I. 2.3–8.4, *p* < 0.001) among the ERA group. Higher ESR (OR 1.03, C.I. 1.002–1.051, *p* = 0.03), CRP (OR 1.03, C.I. 1.002–1.061, *p* = 0.04), lower protein intake (OR 0.98 C.I. 0.96–0.99, *p* = 0.04), corticosteroid usage (OR 3.71 C.I. 1.4–9.9, *p* < 0.01) and lower quality of life (higher HAQ score OR 2.41 C.I. 1.24–4.65, *p* < 0.01) were associated with having low lean mass in the ERA group (adjusted to age and gender). *Conclusions*: Patients with early RA have lower appendicular lean mass and higher body fat percentage compared to healthy controls. Loss of lean mass in early RA is associated with elevated inflammatory markers inducing catabolism, lower protein intake and also with GCS treatment.

## 1. Introduction

Rheumatoid arthritis (RA) is a chronic inflammatory disease associated with changes in body composition [1,2,3,4]. Evidence of muscle wasting and fat mass gain (overfat) without a change in body weight, known as rheumatoid cachexia, have been reported in two thirds of RA patients with established disease [1,2]. Low lean mass (LM) in RA is an important predictor of disability [5,6] and cardiovascular mortality [4,7,8].

The loss of muscle tissue in RA appears to be mediated through pro-inflammatory sarcoactive cytokines promoting proteolysis [1,2,3,9]. In general population diet and physical inactivity are the most important contributors to sarcopenia development, lifestyle factors could contribute to lean mass loss in RA as well [10,11]. Corticosteroid (GCS) treatment through increased protein breakdown and decreased synthesis is another factor possibly involved in both loss of lean mass and fat mass gain in RA [1].

The handful of studies focusing on BC in recent onset RA have revealed that lean mass loss and fat mass gain can be evident already in the initial stage of RA but less is known for which factors contribute to the unfavorable BC [12,13,14]. In a Swedish study early rheumatoid arthritis (ERA) patients had lower appendicular lean mass (ALM) compared to controls, but no association was found with disease specific measures [13]. A higher prevalence of altered BC has been reported in Vietnamese women with ERA compared to matched controls, in this group a change in BC was associated with RA activity and disability, a lower frequency of regular exercise was observed in sarcopenic subjects [12]. BC in treatment naïve older early arthritis patients was evaluated in a recent study by Turk et al., loss of muscle mass was 4–5 *times* more common in early arthritis patients than controls but no associations between disease activity and an unfavorable BC were found [15]. In previous studies the role of lifestyle and RA disease specific parameters in association with lean mass parameters have rarely been assessed.

The aim of our study was to evaluate BC of RA patients at disease onset compared to population controls focusing on the associations between low lean mass and disease specific parameters, nutritional factors and physical activity.

## 2. Materials and Methods

The study group in this cross-sectional study consisted of 91 patients with ERA (aged 19–79y) and 328 control subjects (aged 20–79y).

For the ERA group all consecutive patients referred to a tertiary care center in January 2012 to May 2014 with a first ever RA diagnosis and symptom duration up to one year (early arthritis) were invited to participate in the study. To be included in the study the ACR/ EULAR 2012 classification criteria for RA had to be fulfilled. Patients with other inflammatory joint conditions were excluded, no other exclusion criteria were applied.

To form the control group, subjects adjusted for the age and gender of the general population of the area in 2013 were randomly selected from a primary health care center practice list (total number of subjects 1854). No exclusion criteria were applied. Postal invitations were sent out inviting to contact the primary health care center to participate in the study. All subjects willing to participate provided written informed consent at the health care center, further study activities were performed at the tertiary care center.

Study procedures were carried out after an overnight fast. Body weight was measured in kilograms with a calibrated electronic scale. Height was measured to the nearest 0.5  cm using a stadiometer.

In the RA group, erythrocyte sedimentation rate-ESR was measured using modified Westergren method. Seropositivity for anti-citrullinated protein antibodies (antiCCP) and rheumatoid factor (RF) was evaluated. AntiCCP was measured using electrochemoluminescence-assay, using the value of 17k U/L as the cut-off for positivity. To measure RF immunoturbidimetric method was used and the test was considered positive if RF value was >14 IU/mL. CRP was measured in both of the study groups using immunoturbidimetric method.

The number of tender and swollen joints was recorded in the ERA group (28 and 44 joint scores) and disease activity score DAS28 was calculated accordingly [16].

Subjects were grouped according to body mass index (BMI) values by the WHO criteria [17]–normal weight (BMI ≤ 24.9 kg/h^2^), overweight (BMI 25–29.9 kg/h^2^), and obese (BMI ≥ 30 kg/h^2^).

BC parameters—fat mass, fat-free mass, and ALM were measured with a Lunar Prodigy Advance Dual Energy X-Ray absorptiometry (DXA) machine. Body fat percentage (BFP, the fat percentage of total body mass), and the appendicular lean mass index ALM/h^2^ (appendicular lean mass/height squared) were calculated. As there is no universal definition for low lean mass in RA we defined low lean mass as having an ALM/h^2^ less than the 20th percentile of the sex specific control group values, corresponding to a threshold value of 8.0586 kg/h^2^ for males, and 6.0359 kg/h^2^ for females. Overfat was defined as BFP >25% for men and >35% for women [17]. Using these cut-off values, BC phenotypes were defined: overfat, low lean mass, overfat with low lean mass. A subject was classified as having healthy BC if both lean and fat mass were within normal values.

A 24-h dietary recall (24 HDR) capturing information about foods and beverages consumed in the past 24 h was used to evaluate energy and nutrient intake. The 24HDR method provides detailed intake data but cannot account for day- to day variability and is unable to assess long-term dietary exposure. NutriData software [18] was used to translate foods and beverages into nutrient equivalents. Physical activity was evaluated using the International Physical Activity Questionnaire Short Form (IPAQ-SF) [19,20]. At least 150 min of moderate—or 75 min of vigorous—intensity physical activity throughout the week was considered to be sufficient as recommended by the WHO [21]. The quality of life of RA patients was assessed with the Health Assessment Questionnaire Disability Index (HAQ-DI) questionnaire.

To evaluate differences between the groups, Chi-square test, Fisher’s exact test when the assumptions of chi-square test were not met or Mann-Whitney U test were used as appropriate. Two-tailed tests and a 5% significance level with Bonferroni correction for multiple comparisons were used in all analyses.

Binomial logistic regression was performed to ascertain the effects of age and gender adjusted subject characteristics on the likelihood of low lean mass and the associations between ERA and BC phenotypes. The linearity of the continuous variables with respect to the logit of the dependent variable was assessed via the Box-Tidwell (1962) procedure.

All analyses were carried out using SPSS v24 for Windows.

Availability of data and materials: The datasets used and/or analyzed during the current study are available from the corresponding author on reasonable request.

The study was approved by the Research Ethics Committee of University of Tartu (approval no. 232/M-13, date of approval 03.10.2011). All subjects participating in the study signed written informed consent forms.

## 3. Results

100 patients with newly diagnosed RA were recruited between 2012 and 2014. Six patients did not meet the ACR/EULAR criteria for RA [22]. Two patients with diagnosis of other inflammatory joint condition were excluded-one had ankylosing spondylitis, one arthritis associated with HCV. One patient was excluded due to achondroplasia (n = 91). No patients in the RA group had concomitant pulmonary condition, heart failure or malignancy considered clinically important by the investigator, there was one case of ovarian cancer in medical history.

To form the control group postal invitations were sent out to 350 subjects, 332 contacted the primary health care center and were recruited in October 2014 to March 2015. Three subjects missed their study appointment, and one with missing BC data was excluded (n = 328). The age and gender distribution of the control group subjects used in the data analysis matched the age-sex structure of the general Estonian population [23]. No patients in the control group had clinically significant concomitant pulmonary condition, heart failure or malignancy, there was one case of prostate cancer, two cases of breast cancer in self-reported medical history.

The main characteristics of the study groups are presented in Table 1. Mean age in the ERA group was 52, and 48 years in the control group (*p* = 0.03). 72% of the patients and 54% of the controls were female (*p* < 0.001). The control subjects were taller and heavier than RA patients but there was no difference in the mean BMI value or prevalence of overweight or obesity between the groups.

Patients with ERA consumed fewer total food calories per day (*p* = 0.001) (mean protein intake 23 g lower (*p* < 0.001) and mean fat intake 16 g lower (*p* = 0.001)). 28% of men with ERA reached sufficient physical activity compared to 64.2% of the control group (*p* = 0.01). However, over half (53%) of the women with ERA were sufficiently physically active, as were only 40.7% of the women in the control group (*p* = 0.01).

Patients with ERA had a significantly higher mean CRP value compared to control subjects. The majority of RA patients were seropositive and had moderate disease activity according to the ACR 2012 recommendations [16], the mean DAS28 score was 4.2. Male RA patients had higher disease activity, levels of inflammatory markers, and a higher proportion of antiCCP and RF seropositivity. The mean duration of RA associated symptoms (disease duration) was 215 days, and time from RA diagnosis 36.6 days. 56% of the patients had started disease-modifying therapy and 29% were using glucocorticosteroids (mean dose 14 mg) at the time of the study visit. The duration of therapy was not specified. The disease-specific characteristics of the ERA group are shown in Table 2.

The BC parameters of the ERA and control group are shown in Table 3, the prevalence and age and gender adjusted odds of BC phenotypes in the groups in Table 4. ERA patients had higher mean BFP, lower ALM values and higher odds of having unhealthy BC compared to age and gender adjusted controls. 41.8% of the ERA patients and 19.8% of the controls had low lean mass (adjusted OR 3.3 (95% C.I. 1.9–5.5, *p* < 0.001). 68.1% of the ERA subjects and 47.3% of the controls were overfat (adjusted OR 1.9 (95% C.I. 1.1–3.3, *p* = 0.02)) and the adjusted odds of sarcopenic overfat were 4.4 *times* higher (26.4% vs. 7.0% 95% C.I. 2.3–8.4, *p* < 0.001) among the ERA group. The control subjects had 2.9 *times* higher odds of having healthy BC (95% C.I. 1.5–5.3, *p* = 0.001) than age and gender adjusted patients with ERA.

Higher inflammatory activity measured by ESR (OR 1.03, C.I. 1.002–1.051, *p* = 0.03) and CRP (OR 1.03, C.I. 1.002–1.061, *p* = 0.04), lower protein intake (OR 0.98 C.I. 0.96–0.99, *p* = 0.04), corticosteroid usage (OR 3.71 C.I. 1.4–9.9, *p* < 0.01) and lower quality of life (higher HAQ score) were associated with low lean mass in the ERA group (adjusted to age and gender) as shown in Table 5.

In the control group, the low ALM was associated with age, insufficient physical activity, and smoking.

## 4. Discussion

To our knowledge, this is the first study to evaluate both -the role of lifestyle and disease specific parameters in association with low lean mass in ERA. We found that the BC of patients with recent onset RA differs from control subjects—ERA patients have lower ALM, higher BFP, and a higher prevalence of unhealthy BC phenotypes while there is no difference in BMI between the groups. According to our results low lean mass in ERA is associated with inflammatory activity, GCS usage, and protein intake.

It is known that the majority of patients with RA may be affected BC alteration-lean mass loss and fat mass gain in the established disease [1,2], but there is a lack of comparable results in early ERA. Dao [12] found that 12.4 percent of women with early stage RA were sarcopenic and overfat (had rheumatoid cachexia), but in long-standing disease it has been reported in up to two-thirds of patients [1,2]. 42% of the subjects in our ERA group were classified as having low lean mass and 26% had the least favorable BC phenotype: a combination of low lean mass and overfat. Only 16% of the patients with recent onset RA had no unhealthy BC components. We did not find an association between duration of symptoms indicative to arthritis and low lean mass. This suggests that pre-clinical factors contribute to the change at a greater extent than reduction of physical activity due to pain and stiffness induced by arthritis. Our results differ lightly from those of previous studies-Book [13] found that disease duration was associated with lower lean mass in females in the initial stage of RA but after two years of follow-up [14] concluded that age-related deterioration in lean mass and fat mass gain are smaller in RA compared to controls, Turk [15] found longer symptom duration to be related with a higher ALMI in male subjects only. In light of the knowledge that BC change in RA leads to insulin resistance, metabolic syndrome, endothelial dysfunction, and higher cardiovascular risk, [4,9,15,24,25] the findings in the initial stage of the disease are concerning, but their prognostic value needs to be clarified in future studies.

Our results confirm that the change in BC in RA develops early in the course of the disease [12,13,15] but there seem to be important geographic differences in BC parameters of RA patients making direct comparisons of data difficult. In a Swedish group of patients with recent onset RA similarly lower than control group ALM values were reported (17.2 kg in women and 22.5 kg in men, 17.2 kg and 23.7 kg respectively in our group) [13,14]. When BC was evaluated in Vietnamese women with ERA, the mean ALM was 12.9 kg [12], while a Welsh group reported mean ALM levels of 14.0 kg for women and 21.0 kg for men [26]. In a recent early arthritis study in the Netherlands a 5–7% lower ALM was found in DMARD naïve patients compared to age matched controls, but only data of subjects over the age of 50 was analyzed [15].

Protein degradation through stimulation of catabolism by inflammatory cytokines is considered to be the main mechanism responsible for the development of rheumatoid cachexia characteristic to established disease [1,2,3]. Muscle loss can occur in several other chronic inflammatory conditions such as malignancy, heart failure, sepsis, chronic obstructive pulmonary disease, chronic kidney, liver disease and HIV among others [27]. Biological ageing is characterized by chronic inflammation while higher levels of CRP and IL-6 are considered to be predictors of age-related sarcopenia [28]. In our early RA group lean mass reduction was associated with acute phase response (ESR, CRP), but not with prognostic factors (antiCCP/RF positivity) or indices (DAS28, joint counts) that correlate with burden of the disease and reduction of physical activity as a consequence. The finding leads to a hypothesis that through controlling inflammation, a stabilization in the loss of lean mass could be achieved. Previously a weak association between low lean mass and disease activity was found with DAS28 in one ERA study [13] but not in others [12,15]. However, regarding the effect of disease modifying (DMARD) treatment on BC, the results have been contradictory. Tournadre reported an increase in lean mass after 1 year of treatment with an IL-6 inhibitor [29]. Other groups have found that through achieving remission using tightly controlled therapy, the alteration in BC could be stabilized [14] but the lean mass loss cannot be reversed even if the treatment is started early [26,30]. A likely explanation is that the change in BC occurs very early in the course of RA, probably in the preclinical phase [31] and is resistant to alteration without specific pro-anabolic interventions.

While DMARD treatment can potentially stabilize BC alteration the long-term effect of GCS usage leading to muscle wasting, fat accumulation, and fat redistribution, is a well-known phenomenon [32,33]. We found the current GCS usage to be associated with 3.7 *times* higher odds of having low lean mass (C.I. 1.4–9.9, *p* = 0.009) among the ERA subjects but no association with DMARD therapy was seen. The main mechanism responsible for GCS induced muscle loss results from increased protein breakdown and decreased synthesis [32]. There have not been many reports on the short-term effects of GCS use. Dao et al. did not find GCS usage to be associated with a BC change in their ERA study [12], but in 2016, in a group of ERA patients treated with high-dose, step-down GCS regimens, an increase in FM without lean mass reduction was found, [34] suggesting that a rapid decrease in inflammation counteracted the negative effects on BC. Interestingly in one study loss of lean mass was reported after a single high-dose intramuscular GCS injection [35]. In our group, an association with low lean mass appeared after a short-term, medium dose GCS use: the mean dose of prednisolone was 13.8 mg and mean time from diagnosis 37 days.

In addition to disease specific features, we studied the associations between lifestyle factors and low lean mass. Healthy diet, especially sufficient protein intake is essential in the prevention of age-related loss of muscle mass, and protein supplementation is recommended for treating sarcopenia in the elderly population [10,11,36]. Previously it has been suggested that RA patients are not undernourished, and have protein intake comparable to healthy controls [1,4,37]. Using the 24 h dietary recall we found the total calories consumed, fat and protein intake of ERA patients to be lower than of the control subjects and low protein intake was associated with the presence of low lean mass (adjusted to age and gender). There is a lack of comparable data on nutrition in the development of BC changes in ERA but there is some evidence that dietary change may provide benefits in reducing pain and swollen and tender joints [33,38,39]. We could only find one study estimating the role of nutrition in muscle loss associated with RA, no association was found between protein intake and low lean mass in this group of established RA patients [40]. Our finding suggests that nutritional advice may have a valuable role in RA patient education, and the potential benefit of nutrition, and protein supplementation to gain lean mass, is a topic to be looked into in future research.

Low muscle mass could be the consequence of lack of exercise, maintaining sufficient physical activity is the cornerstone in the prevention of age related muscle loss [10,11], significantly less is known about RA. The majority of RA patients tend to be physically inactive, the main factors limiting activity are disuse due to arthralgia and stiffness, and active inflammation inducing further muscle loss [1,41]. Lower frequency of regular intentional exercise was observed in a group of female sarcopenic early RA subjects by Dao in 2011 [12]. High intensity resistance training has been shown to improve BC in RA [1,42,43]. In our group, there was a notable gender difference in physical activity-women with ERA were more active than control women, but the opposite was found in men. We found the low ALM to be associated with physical inactivity among control subjects, but we did not find an association between physical activity and low lean mass in the ERA group. This finding in the light of the association between elevated inflammatory markers and low lean mass emphasizes that hypermetabolism caused by inflammation [1] is more important in the loss of muscle tissue in RA than physical inactivity: the main determinant of low lean mass among the population controls.

Finally in accordance with previous studies [5,8,12,13] in our early RA group disability was associated with higher probability of having low lean mass. A 2.4 *times* increase in odds for low lean mass was seen per every unit increase in HAQ score confirming the association between quality of life and muscle loss.

Several limitations to our study should be acknowledged. The sample size of the ERA group and small number of male subjects limited possibilities in statistical analysis. The difference in age and gender composition between RA subjects and controls did not allow direct comparison between the groups, age and gender adjustment was used in data analysis to limit the bias. As the change in BC seems to appear in the earliest stage of the disease, the lack of pre-treatment data on BC, disease activity, and lifestyle factors can also considered to be a limitation. The presence of sarcopenia—a combination of low muscle mass, inadequate strength and low performance is more important in health outcomes than muscle mass alone. For defining sarcopenia according to the current EWGSOP criteria originally designed to be used in elderly population muscle strength and performance have to be taken into account for [44]. Measures of muscle strength (e.g., grip strength) in patients with RA can be affected by various patient-related factors such as pain, arthritis activity, deformities and disability resulting in values lower than in healthy individuals [45,46]. As there are no validated criteria for sarcopenia in RA, we defined low muscle mass by lower 20th percentile of gender—specific appendicular lean mass values.

## 5. Conclusions

A change in body composition appears in the initial stage of RA. Patients with early RA have lower appendicular lean mass and higher body fat percentage compared to healthy controls. Loss of lean mass in early RA is associated with elevated inflammatory markers inducing catabolism, lower protein intake and also with GCS treatment. The results indicate that targeting BC through adequate control of inflammation, nutritional advice and minimizing the use of GCS might have a valuable role in future management strategies of RA.

## Figures and Tables

**Table 1 medicina-55-00730-t001:** Subject characteristics by group and gender.

	Early RA N = 91	Control Group N = 328	Male		Female	
			Early RA N = 25	Control Group N = 151	Early RA N = 66	Control Group N = 177
Gender (female)	66 (72) ^a^	177 (54) ^a^				
Age (years)	52 (2) ^a^	48 (1) ^a^	55 (3) ^c^	46 (1) ^c^	51 (2)	50 (1)
Smoking (ever)	30 (33.0) ^b^	67 (20.0) ^b^	15 (60.0) ^c^	31 (20.5) ^c^	15 (22.7)	3 (20.3)
CRP (mg/dL)	12.3 (2.0) ^c^	2.3 (0.2) ^c^	22.5 (5.3) ^c^	1.9 (0.3) ^c^	8.4 (1.7) ^c^	2.6 (0.3) ^c^
Height (cm)	1.67 (0.01) ^c^	1.72 (0.01) ^c^	1.75 (0.01) ^c^	1.80 (0.01) ^c^	1.64 (0.01)	1.65 (0.01)
Weight (kg)	74.8 (1.5) ^b^	80.4 (1.0) ^b^	80.5 (2.5) ^b^	89.2 (1.3) ^b^	72.6 (1.8)	73.0 (1.2)
BMI (kg/h^2^)	27.1 (0.6)	27.3 (0.3)	26.5 (0.9)	27.6 (0.4)	27.3 (0.7)	27.0 (0.5)
Normal weight N (%)	37 (40.7)	124 (37.8)	10 (40.0)	49 (32.5)	27 (40.9)	75 (42.4)
Overweight	31 (34.1)	107 (32.6)	10 (40.0)	56 (37.1)	21 (31.8)	51 (28.8)
Obese N (%)	23 (25.3)	97 (29.6)	5 (20.0)	46 (30.5)	18 (27.3)	51 (28.8)
Abdominal obesity N (%)	43 (47.3)	142 (43.3)	8 (32.0)	57 (37.7)	35 (53.0)	85 (48.0)
Nutrient intake/24 h						
Kcal	1534 (69) ^c^	1841 (44) ^c^	1695 (122) ^a^	2086 (76) ^a^	1471 (83)	1631(43)
Carbohydrates (g)	179 (10)	191 (5)	193 (14)	209 (7)	174 (13)	175 (6)
Fat (g)	59 (3) ^c^	75 (2) ^c^	67 (5)	86 (4)	56 (3) ^a^	66 (2) ^a^
Protein (g)	57 (3) ^c^	80 (2) ^c^	62 (6) ^b^	93 (4) ^b^	55 (3) ^c^	69 (2) ^c^
Physical activity						
Moderate/vigorous (min/week)	383.1 (56.9)	299.7 (23.5)	316.0 (110.0)	372.0 (35.0)	406.3 (66.7) ^a^	237.8 (31.0) ^a^
Sufficient N (%)	42 (46.2)	170 (51.8)	7 (28.0) ^b^	97 (64.2) ^b^	35 (53.0) ^b^	73 (41.2) ^b^

Mean, ± S.E. if not stated otherwise. ^a^
*p* < 0.05, ^b^
*p* < 0.01, ^c^
*p* < 0.001 CRP: C-reactive protein, BMI-body mass index.

**Table 2 medicina-55-00730-t002:** Disease activity and characteristics of patients with rheumatoid arthritis.

	Male N = 25	Female N = 66	Total N = 91	*p*-Value
Time from first symptoms (days)	201 (58)	220 (25)	215 (24)	NS
Time from RA diagnosis (days)	26.8 (10.3)	40.2 (5.7)	36.6 (5.0)	NS
ESR mm/h	35.0 (5.6)	14.7 (1.9)	20.3 (2.3)	<0.001
CRP mg/L	22.5 (5.3)	8.4 (1.7)	12.3 (2.0)	<0.001
ACPA positive N (%)	22 (88.0)	41 (62.1)	63 (69.2)	0.02
RF positive N (%)	22 (88.0)	42 (63.6)	64 (70.3)	0.02
Swollen joint count (N/44)	6.4 (1.4)	4.2 (0.5)	4.8 (0.6)	NS
Tender joint count (N/44)	9.7 (1.6)	10.9 (1.0)	10.6 (0.9)	NS
DAS 28 score	4.8 (0.3)	3.9 (0.2)	4.1 (0.2)	0.02
Using DMARD N (%)	11 (44.0)	40 (60.6)	51 (56.0)	NS
Using GCS N (%)	9 (36.0)	17 (25.8)	26 (28.6)	NS
GCS dose (mg)	13.3 (1.9)	14.1 (1.5)	13.8 (1.2)	NS

Mean, ± S.E. if not stated otherwise. Statistically significant differences between the male and female subjects (*p* < 0.05). RF: Rheumatoid Factor; DAS28: Disease Activity Score calculated using 28 joints; GCS: Glucocorticosteroids; DMARD: Disease-Modifying Anti-Rheumatic Drugs; Low disease activity: (DAS 28 score < 3.2); moderate disease activity (DAS28 ≥ 3.2 to ≃ 5.1); high disease activity (DAS 28 > 5.1). High CRP-CRP > 10mg/L, high ESR-ESR > 15 mm/h.

**Table 3 medicina-55-00730-t003:** Body composition in RA patients and controls.

	Early RA N = 91	Control Group N = 328		Male			Female		
			*p*	Early RA N = 25	Control Group N = 151	*p*	Early RA N = 66	Control Group N = 171	*p*
Fat mass (kg)	27.3 (1.1)	26.5 (0.6)	NS	22.5 (1.9)	24.1 (0.9)	NS	29.2 (1.3)	28.5 (0.9)	NS
BFP	36.0 (1.0)	29.4 (0.6)	<0.001	27.2 (1.7)	21.9 (0.6)	0.002	39.3 (0.9)	35.6 (0.7)	0.007
Trunk fat %	53.6 (0.7)	55.8 (0.4)	0.01	59.3 (1.2)	60.7 (0.5)	0.01	51.5 (0.7)	51.6 (0.4)	NS
Appendicular fat %	43.1 (0.7)	40.9 (0.4)	0.01	37.0 (1.1)	35.9 (0.4)	NS	45.4 (0.7)	45.2 (0.4)	NS
ALM (kg)	19.0 (0.4)	22.9 (0.3)	<0.001	23.7 (0.6)	28.5 (0.3)	<0.001	17.2 (0.3)	18.1 (0.2)	0.01
ALM/h^2^	6.8 (0.1)	7.7 (0.1)	<0.001	7.7 (0.2)	8.8 (0.1)	<0.001	6.4 (0.1)	6.7 (0.1)	0.03

Values are the mean, ± S.E. Statistically significant differences *p* < 0.05. BFP: Body Fat Percentage (fat % of total body mass); trunk fat %: percentage of trunk fat of total fat; appendicular fat %: percentage of appendicular fat of total fat; ALM: Appendicular Lean Mass; ALM/h^2^: ALM /height^2^).

**Table 4 medicina-55-00730-t004:** Prevalence and age, gender adjusted odds for BC phenotypes.

	Early RA N = 91		Control Group N = 328				
	N (%)	95% C.I,	N (%)	95% C.I.	*p*	aOR (95% C.I.)	*p*
Healthy BC	15 (16.5)	10.0–25.1	130 (39.6)	34.5–45.0	<0.001	0.4 (0.2–0.7)	0.01
Low lean mass	38 (41.8)	32.0–52.0	65 (19.8)	15.8–24.4	<0.001	3.3 (1.9–5.5)	<0.001
Overfat	62 (68.1)	58.1–77.0	155 (47.3)	41.9–52.7	<0.001	1.9 (1.1–3.3)	
Low lean mass+ overfat	24 (26.4)	18.2–36.1	23 (7.0)	4.6–10.2	<0.001	4.4 (2.3–8.4)	<0.001

Age, gender adjusted odds ratio (aOR) for early RA group (binary logistic regression) healthy BC: normal lean mass, normal fat mass; low lean mass: low appendicular lean mass; overfat: high body fat percentage.

**Table 5 medicina-55-00730-t005:** Factors associated with low lean mass in the early rheumatoid arthritis and control group adjusted for age and gender.

	Early RA				Control Group			
	aOR	95% C.I.		*p*	aOR	95% C.I.		*p*
Age (years) *	1.00	0.98	1.03	0.77	0.98	0.96	0.99	0.006
Gender (male) *	2.23	0.87	5.68	0.09	1.01	0.59	1.75	0.96
ESR (mm/h)	1.026	1.002	1.051	0.03	ND			
CRP (mg/L)	1.032	1.002	1.063	0.04	0.99	0.91	1.07	0.74
HAQ score (units)	2.41	1.24	4.65	0.009	ND			
25 (OH) vitamin D (n mol/L)	1.00	0.98	1.02	0.84	0.99	0.98	1.01	0.30
Tender joint count (N/44)	1.02	0.97	1.08	0.39	ND			
Swollen joint count (N/44)	1.03	0.95	1.12	0.53	ND			
DAS 28 score (units)	1.22	0.91	1.63	0.18	ND			
RF positive	1.10	0.42	2.88	0.84	ND			
ACPA positive	1.20	0.46	3.13	0.71	ND			
Current GCS user	3.71	1.39	9.94	0.009	ND			
Current DMARD user	1.29	0.54	3.09	0.57	ND			
Time from first symptoms	1.00	1.00	1.00	0.55	ND			
Insufficient physical activity	0.51	1.00	1.34	0.17	2.99	1.64	5.46	<0.001
Smoking (ever)	1.40	0.53	3.67	0.50	2.35	1.27	4.33	0.006
Protein intake (g/day)	0.98	0.96	0.99	0.04	1.00	0.99	1.00	0.43

aOR: age, gender adjusted odds ratio (binary logistic regression) *: Unadjusted; HAQ score: Health Assessment Questionnaire; DAS 28: Disease Assessment Score 28; RF: Rheumatoid Factor; ACPA: Anti-cyclic Citrullinated Peptide Antibodies; GCS: glucocorticosteroid; DMARD: Disease-Modifying Anti-Rheumatic Drug; sufficient physical activity: >150 min moderate/75 min vigorous activity/week; ND-not done.

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
