# Peer review of "Factors Associated with Low Lean Mass in Early Rheumatoid Arthritis: A Cross-Sectional Study"

_medicina, 2019, doi:10.3390/medicina55110730_

Round 1

Reviewer 1 Report

The authors revealed the factors associated with low lean mass in RA. Considering the influence to ability of daily living, low lean mass is important for patients. So, the point of focus is interesting, but several numbers of concerns need to be addressed by authors. 

Various diseases such as malignancy, heart failure, respiratory disease influence to body lean mass/ body fat. Both comorbidity and past history are important, do the patients who enrolled this study have no comorbidity or past history of such disease? If not, the patients who have such comorbidity should be excluded. Regarding about characteristics of patients, the duration of GC use should be described. Overall, the disease duration of patients who are enrolled in this study was extremely low (about one month). Considering this, the RA seemed to less contribute to body lean mass. The authors should discuss from this point of view. And I kindly recommend the author should analyze in established-RA patients in next study. The disease activity was not related with low lean mass, but inflammation was related with low lean mass. How about other inflammatory disease such as Castleman disease, other autoimmune diseases? If the knowledge about the relation with other inflammatory diseases are present the author should discuss.

Author Response

Authors’ response to the reviewers’ comments:

Thank you for you thorough review, several important points were noticed by the reviewer that will improve the manuscript. Each comment has been carefully considered and responded. Responses to the reviewer and changes in the revised manuscript are as follows. Changes in the manuscript are highlighted in yellow.

Response to Reviewer 1 comments:

The authors revealed the factors associated with low lean mass in RA. Considering the influence to ability of daily living, low lean mass is important for patients. So, the point of focus is interesting, but several numbers of concerns need to be addressed by authors. 

Point1

Various diseases such as malignancy, heart failure, respiratory disease influence to body lean mass/ body fat. Both comorbidity and past history are important, do the patients who enrolled this study have no comorbidity or past history of such disease? If not, the patients who have such comorbidity should be excluded.

Response1:

Thank you for this important note. Data on comorbidity were collected but no exclusion criteria were applied for malignancy, heart failure or chronic respiratory disease in the early RA or control group. No subjects in either of the groups had clinically significant pulmonary condition or heart failure. Data on comorbidities in the RA group (lines 122-124) and control subjects (lines 129-131) have been added to the manuscript.

Point 2

Regarding about characteristics of patients, the duration of GC use should be described. Overall, the disease duration of patients who are enrolled in this study was extremely low (about one month). Considering this, the RA seemed to less contribute to body lean mass. The authors should discuss from this point of view. And I kindly recommend the author should analyze in established-RA patients in next study.

Response 2.

We agree with the reviewer that glucocorticosteroid therapy is an important point to be considered- unfortunately the duration of GCS treatment was not specified (clarification added in lines 149-151). As the time from diagnosis is ~37 days we can only speculate that this is the mean duration of GCS treatment as well. Regarding disease duration- the mean duration of symptoms indicative to arthritis was 215days (please see table 2- line 152), time from diagnosis about one month showing an important delay in the diagnosis. Duration of symptoms was not associated with low lean mass (miswording corrected in table 5), discussion added to lines 202-209

Thank you for the recommendation to analyze body composition in established RA. This is our aim in next stages of our research as a follow- up to this study and in future projects.

Point3

The disease activity was not related with low lean mass, but inflammation was related with low lean mass. How about other inflammatory disease such as Castleman disease, other autoimmune diseases? If the knowledge about the relation with other inflammatory diseases are present the author should discuss.

Response 3

We agree with the reviewer that the role of inflammation deserves further attention, discussion on the topic (inflammation in age related loss of lean mass and other inflammatory conditions) has been added to lines 222-227. We could not find any studies on Castleman disease and loss of lean mass.

Short clarification on disease activity parameters and inflammatory markers added to lines 229-230 and 276- 277

Reviewer 2 Report

Methods: line 61 to 76 - seems to be more results than methods. Outcomes and research process is confuse.

Results: a Flow chart should be include

Author Response

Authors’ response to the reviewers’ comments:

Thank you for you review, several important points were noticed by the reviewer that will improve the manuscript. Each comment has been carefully considered and responded. Responses to the reviewer and changes in the revised manuscript are as follows. Changes in the manuscript are marked in yellow.

Response to Reviewer 2 comments

Comments and Suggestions for Authors

Point 1

Methods: line 61 to 76 - seems to be more results than methods. Outcomes and research process is confuse

Response 1

We agree with the reviewer- the detailed data on participants belongs to the results section. We have restructured these sections as per reviewer's advice (please see lines 63- 73 in material and methods and 119- 131 in results)

Point 2

Results: a Flow chart should be include

Response 2

Thank you for the suggestion, we have written out the details on how the final groups of subjects participating in the study were formed. In this manuscript we decided not to include a flow chart but in future work will keep the suggestion in mind when reporting the results of the follow- up study.

Round 2

Reviewer 1 Report

The author replied appropriately and revised manuscript has been improved.